# Binding and structural basis of equine ACE2 to RBDs from SARS-CoV, SARS-CoV-2 and related coronaviruses

Zepeng Xu[1,2,5], Xinrui Kang[1,3,5], Pu Han[1,5], Pei Du[1,5], Linjie Li[1,4], Anqi Zheng[1,4], Chuxia Deng[2], Jianxun Qi [1,4], Xin Zhao [1], Qihui Wang [1,4✉], Kefang Liu[1✉] & George Fu Gao [1]

The origin and host range of SARS-CoV-2, the causative agent of coronavirus disease 2019 (COVID-19), are important scientific questions as they might provide insight into understanding of the potential future spillover to infect humans. Here, we tested the binding between equine angiotensin converting enzyme 2 (eqACE2) and the receptor binding domains (RBDs) of SARS-CoV, SARS-CoV-2 prototype (PT) and variant of concerns (VOCs), as well as their close relatives bat-origin coronavirus (CoV) RaTG13 and pangolin-origin CoVs GX/P2V/2017 and GD/1/2019. We also determined the crystal structures of eqACE2/ RaTG13-RBD, eqACE2/SARS-CoV-2 PT-RBD and eqACE2/Omicron BA.1-RBD. We identified S494 of SARS-COV-2 PT-RBD as an important residue in the eqACE2/SARS-COV-2 PT-RBD interaction and found that N501Y, the commonly recognized enhancing mutation, attenuated the binding affinity with eqACE2. Our work demonstrates that horses are potential targets for SARS-CoV-2 and highlights the importance of continuous surveillance on SARS-CoV-2 and related CoVs to prevent spillover events.

[1] CAS Key Laboratory of Pathogenic Microbiology and Immunology, Institute of Microbiology, Chinese Academy of Sciences, Beijing 100101, China. [2] Faculty of Health Sciences, University of Macau, Macau SAR 999078, China. [3] Savaid Medical School, University of Chinese Academy of Sciences, Beijing 100049, China. [4] University of Chinese Academy of Sciences, Beijing 100049, China. [5] These authors contributed equally: Zepeng Xu, Xinrui Kang, Pu Han, Pei Du. ✉email: wangqihui@im.ac.cn; Liukf@im.ac.cn

SARS-CoV-2, the casing agent of COVID-19[1], can be transmitted from humans to several animal species[2]. Antibodies reactivity against SARS-CoV-2 has been detected in cats and dogs from both Wuhan, China[3] and Italy[4]. Several cats and dogs living in the same household with COVID-19 patients have been tested positive by RT-PCR, indicating that these animals were infected by humans[5,6]. Similarly, SARS-CoV-2-positive minks were reported in farms from multiple countries. The source of the SARS-CoV-2 outbreak in minks was linked to the infected farmers. More importantly, mink-to-human transmission was confirmed by phylogenetic comparison[7,8]. Recently, in several states of the US, a high proportion of white-tailed deer were detected as SARS-CoV-2 positive, and SARS-CoV-2-binding antibodies were also detected in deer serum samples in 2019[9]. Thus, strategies to mitigate the risk of zoonotic SARS-CoV-2 infections have to be developed.

A number of bat-origin coronaviruses (CoVs) have been reported that are closely related to SARS-CoV-2, among which RaTG13, sequenced from *Rhinolophus affinis* (intermediate horseshoe bat) in Yunnan Province[10–12], shares 96.2% genome identity and 90.13% amino acid identity in its receptor binding domain (RBD) of spike (S) protein compared to SARS-CoV-2. In addition, two pangolin-origin CoVs, GD/1/2019 and GX/P2V/2017, display high amino acid similarity in their RBDs to the SARS-CoV-2-RBD (96.9% and 86.6%, respectively). Notably, pangolins carrying the SARS-CoV-2-like CoVs manifest clinical symptoms and histological changes[13,14].

Receptor binding is a prerequisite for virus infection and transmission[15–17]. Angiotensin converting enzyme 2 (ACE2) mediates the entry of SARS-CoV, SARS-CoV-2, GX/P2V/2017, GD/1/2019 and RaTG13 by binding to their RBDs[18]. These five CoVs have broad and overlapping potential host ranges including horses, which, as indicated by previous studies, bind to RBDs from all the five CoVs[18–22]. Notably, RaTG13-RBD displays the highest binding affinity with equine ACE2 (eqACE2) among various species and induces the highest infection rate when challenging various ACE2-expressing cells[18,21]. Indeed, horses are natural hosts for many CoVs[23,24]. Prevalence studies indicated equine CoVs is a co-infecting agent during outbreaks in racecourses in Japan and breeding farms in the US[25–28]. Considering the possibility of recombination between various CoV lineages in the same host and its role in viral evolution[29,30], there is significant risk that recombination occurring among horse-infecting CoVs may further lead to interspecies leaks or even give birth to novel CoVs.

SARS-CoV-2 keeps evolving into new variants in the global transmission[31], among which those with increased transmissibility, severe disease and a higher risk of eluding immunity or testing are classified as variants of concern (VOCs) (https://www.cdc.gov/coronavirus/2019-ncov/variants/variant-info.html). The fifth VOC, Omicron variant, demonstrates higher transmissibility and evoked another wave of global infection[32,33]. Genomic analysis showed that the Omicron BA.1 variant, particularly its RBD, are heavily mutated[34]. Moreover, the Omicron BA.1-RBD contains mutations in all the identified major sites determining the host range of SARS-CoV2 (site 493, 498 and 501)[2], which raises concerns about altered host range of Omicron BA.1-RBD. The identification of potential animal host of Omicron BA.1-RBD is urgently needed.

Currently, the S protein structures of SARS-CoV, SARS-CoV-2 and RaTG13 have been determined[35]. We and others have also characterized the structure of the RaTG13 RBD in complex with human ACE2 (hACE2), as well as the SARS-CoV-2-RBD in complex with ACE2 orthologs from human, cat, bat and pangolin[19,36–39]. Structural analyses show that SARS-CoV-2 and RaTG13 share a similar binding mode, but RaTG13-RBD has

substantially less interactions with hACE2 than that of SARS-CoV-2 RBD[21]. The cryo-EM structures of hACE2 in complex with the RBDs of GX/P2V/2017 and GD/1/2019 have also been reported by our group[40]. We and others have reported the structures of RBDs from SARS-CoV-2 VOCs complexed with hACE2 (particularly Omicron BA.1 variant)[34,41–43], and the results showed that the heavy mutations bring new characteristics. However, no structures of Omicron BA.1-RBD in complex with ACE2 from potential animal hosts has been reported yet.

In this study, we find that eqACE2 broadly binds to the RBDs of SARS-CoV, SARS-CoV-2 prototype (PT), pangolin-origin GX/P2V/2017 and GD/1/2019, bat-origin RaTG13 CoVs as well as SARS-CoV-2 VOCs. We determine the crystal structures of eqACE2 in complex with RaTG13-RBD, SARS-CoV-2 PT-RBD and Omicron BA.1-RBD. The S494 of SARS-COV-2 PT-RBD is identified as a key residue in the eqACE2/SARS-COV-2 PT-RBD interaction. Additionally, we find that N501Y, the commonly recognized enhancing mutation when binding to human, dog and mouse ACE2s, exerts a decreasing effect on binding affinity with eqACE2. These findings indicate that horses are potential targets for SARS-CoV-2 infection and highlight the importance of continuous surveillance on SARS-CoV-2 and its related CoVs, as suggested in our previous studies[2,39].

## Results

### The binding of eqACE2 to the RBDs from SARS-CoV, SARS-CoV-2, GX/P2V/2017, GD/1/2019, RaTG13 and SARS-CoV-2 VOCs.
eqACE2 shares 86.78% amino acid identity with hACE2 and 87.35% in the peptidase domain (19-615) (Supplementary Fig. 1a). To verify the binding affinity of eqACE2 to SARS-CoV, SARS-CoV-2 PT, GX/P2V/2017, GD/1/2019 and RaTG13, eqACE2 and hACE2 were refolded from *Escherichia coli* inclusion bodies, and the five RBDs were expressed by Expi293F cells. Surface plasmon resonance (SPR) was then performed to characterize their binding affinities. Both eqACE2 and hACE2 bound to the five RBDs with a high affinity. The SARS-CoV-2 PT, GX/P2V/2017 and GD/1/2019 RBDs displayed similar affinities among each other when bound to either hACE2 or eqACE2. However, the binding affinity between eqACE2 and the SARS-CoV-RBD was significantly lower than that between hACE2 and the SARS-CoV-RBD (~40-fold). The binding affinity between eqACE2 and the RaTG13-RBD was ~2-fold greater than that between hACE2 and the RaTG13-RBD (Fig. 1a). Sequence alignments demonstrated that the five RBDs are highly conserved but also highlighted diversity at known key sites, such as site 493 and 498[2] (Supplementary Fig. 1b).

Next, we tested the binding affinity of eqACE2 with SARS-CoV-2 PT and VOCs (Fig. 1b and Supplementary Fig. 1c). It's worth noting that the protocol of VOCs' affinity testing is different from the SARS-CoV-2 related CoVs, and the binding affinity can't be directly compared. The binding affinity of Alpha-, Gamma- and Omicron BA.1-RBD displayed extensive decrease with a change of order of magnitude, whereas Beta- and Delta-RBD showed a milder decrease of ~4.1- and ~2.0-fold (Fig. 1b)

### Overall structures of the eqACE2/RaTG13-RBD, eqACE2/SARS-COV-2 PT-RBD and eqACE2/Omicron BA.1-RBD complexes.
To analyze the molecular mechanism of eqACE2 interaction with the SARS-COV-2 PT-RBD, RaTG13-RBD and Omicron BA.1-RBD, the eqACE2/SARS-COV-2 PT-RBD, eqACE2/RaTG13-RBD and eqACE2/Omicron BA.1-RBD complexes were purified (Supplementary Fig. 2). The structures of the eqACE2/RaTG13-RBD, eqACE2/SARS-COV-2 PT-RBD and eqACE2/Omicron BA.1-RBD complexes were determined with a resolution of 2.60 Å, 2.56 Å and 2.86 Å, respectively (Supplementary Table 1).

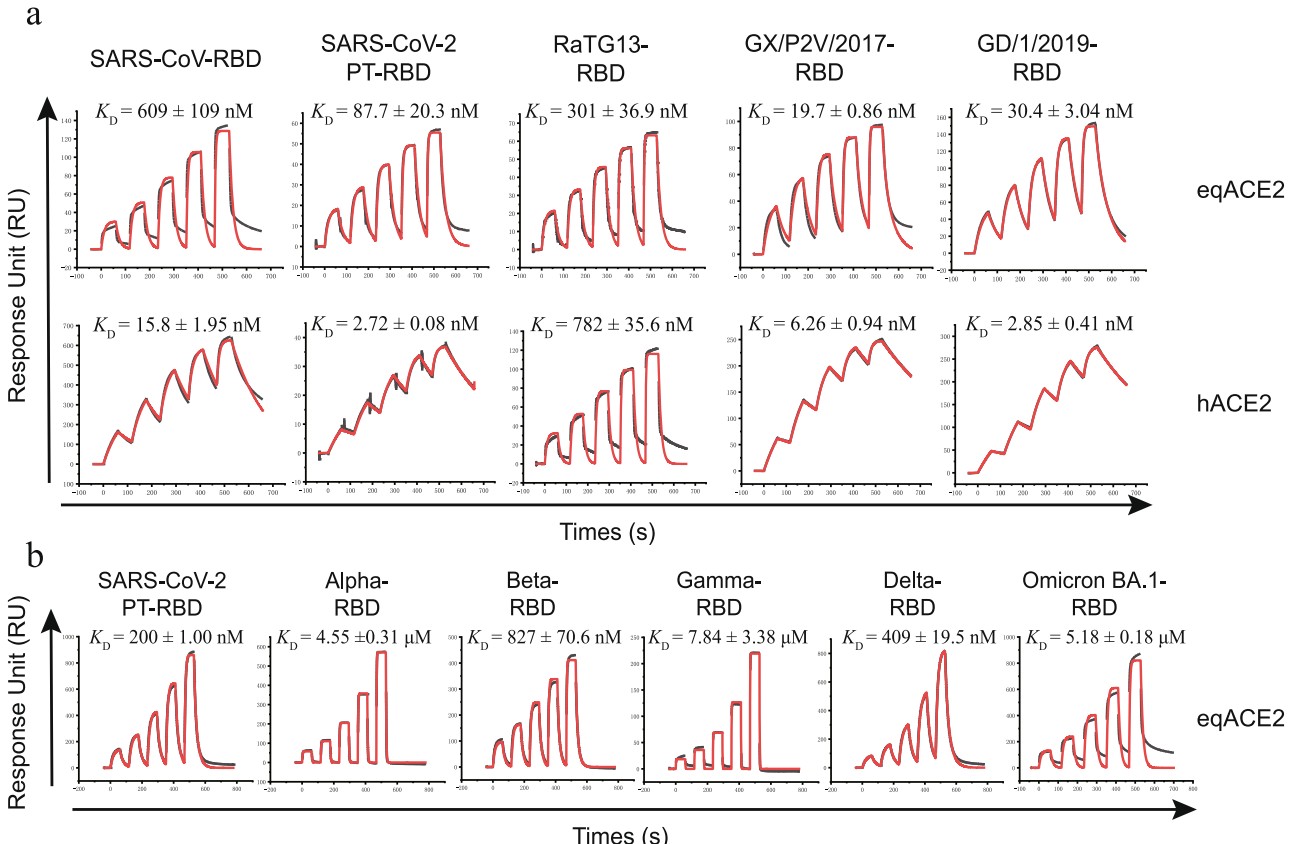

**Fig. 1 Binding eqACE2 to five different coronavirus RBDs and VOCs. a** SPR characterization of the RBDs from SARS-CoV, SARS-CoV-2 PT, RaTG13, GX/ P2V/2017 and GD/1/2019 interacting with equine or hACE2. **b** SPR characterization of the RBDs from SARS-CoV-2 PT and VOCs. Actual and fitted curves are colored in black and red, respectively. The dissociation constant ($K_D$) of each binding test is calculated from three independent repeats and are presented as mean ± SD. Source data are provided as a Source Data file.

In the eqACE2/RaTG13-RBD complex, electron densities for 596 residues of eqACE2 (S19 to A614) and 196 residues of the RaTG13-RBD (N334 to K529), as well as the N-glycan linked to N343 of the RBD, were clearly observed (Fig. 2a). Electron densities of the eqACE2/SARS-COV-2 PT-RBD structure could be observed for 597 residues of eqACE2 (S19 to H615) and 188 residues of the SARS-COV-2 PT-RBD (C336 to T523). There was also an N-glycan on N343 of SARS-CoV-2 PT-RBD (Fig. 2a). Overall, the eqACE2/ RaTG13-RBD and eqACE2/SARS-CoV-2 PT-RBD complexes share a similar architecture, with a root mean squared deviation (RMSD) of 0.424 for 661 Cα atoms (Supplementary Fig. 3a). In the eqACE2/ RaTG13-RBD complex, the RaTG13-RBD, like other beta-CoVs, is composed of two subdomains, in which the external subdomain is dominated by a loop between two small β strands. The eqACE2, like hACE2, can also be divided into two subdomains, among which domain I contacts the RaTG13-RBD (Fig. 2a). The eqACE2/SARS-COV-2 PT-RBD complex displays a conserved architecture compared to hACE2/SARS-COV-2 PT-RBD, with a RMSD of 0.649 for 701 Cα atoms (Supplementary Fig. 3b), but the eqACE2/RaTG13-RBD complex shows more significant divergence with hACE2/RaTG13-RBD (RMSD = 1.443 for 732 Cα atoms)[21]. Alignment between the RBDs of the two complexes demonstrated that RaTG13-RBD binds to eqACE2 at a different angle than to hACE2 (Supplementary Fig. 3c).

Similar to many betaCoV RBDs binding to ACE2 orthologs[21], the 18 RaTG13-RBD residues interacting with eqACE2 are clustered into 2 patches, forming 207 atomic contacts with 20 residues on eqACE2 (Supplementary Table 2). In Patch 1, K417 of the RaTG13-RBD forms a salt-bridge with eqACE2 E30, while N487 forms an H-bond with Y83, L486 and Y489 respectively

forms an H-bond with Y83 and A475 forms an H-bond with S19 (Fig. 2b and Supplementary Table 2). In Patch 2, G496, Y498, D501, T500 and G502 in the RaTG13-RBD form a H-bond network with E38, H41, Q42, K353 and D355 (Fig. 2c and Supplementary Table 2).

The eqACE2/SARS-COV-2 PT-RBD complex shares a similar binding mode with eqACE2/RaTG13-RBD, with 20 residues in the RBD forming 234 atomic contacts with 20 residues in eqACE2 (Supplementary Table 2). However, there are only 10 polar interactions between eqACE2 and SARS-COV-2 PT-RBD, less than the 11 polar interactions between eqACE2 and RaTG13-RBD (Fig. 2b, c, Supplementary Table 2). Notably, S494 on the SARS-COV-2 PT-RBD forms two H-bonds and six van der Waals (vdw) contacts with E38, thus linking the previously reported two patches into one continuous region (Fig. 3d). Besides, an H-bond network has been observed among Y449, Q498 of SARS-COV-2 PT-RBD and E38, K353 of eqACE2 through a water molecule (Fig. 2c).

The molecular interaction of eqACE2/Omicron BA.1-RBD is significantly fewer than hACE2/Omicron BA.1 RBD (244 vs. 287 vdw contacts and 9 vs. 15 polar interactions). Y449 and R498 of Omicron BA.1-RBD form an H-bond network with Q42 and E38 of eqACE2 (Fig. 2c). In contrast to eqACE2/SARS-COV-2 PT-RBD, S494 is not involved in the interaction.

We analyzed differences in the interface residues of the four complexes: eqACE2/RaTG13-RBD, eqACE2/SARS-COV-2 PT-RBD, hACE2/RaTG13-RBD and hACE2/SARS-COV-2 PT-RBD (Fig. 3a–d). There are seven substitutions between the RaTG13-RBD and SARS-COV-2 PT-RBD in complex with eqACE2 (Fig. 3e), among which six have been reported between the two

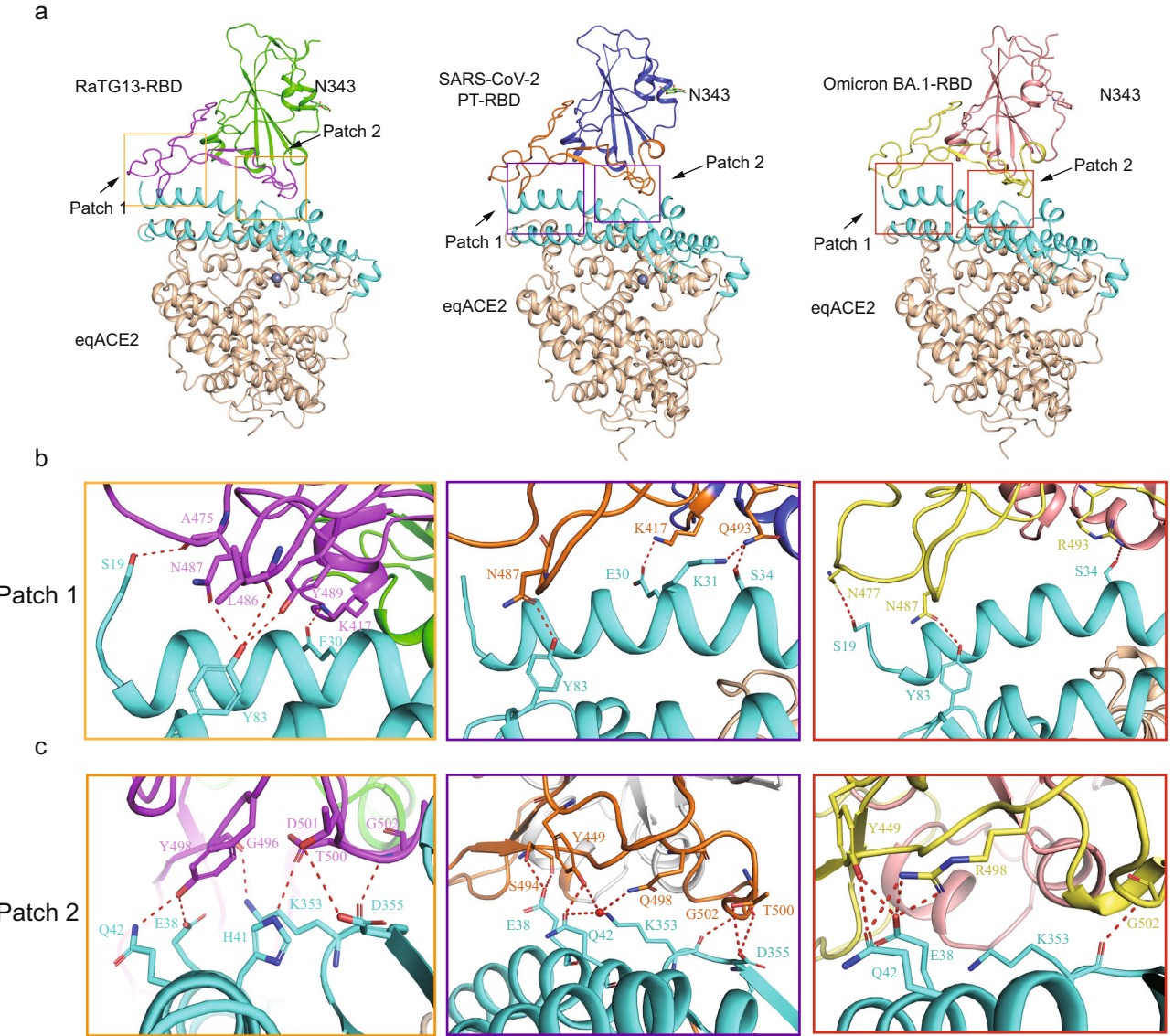

**Fig. 2 Overall architecture of the eqACE2/RaTG13-RBD, eqACE2/SARS-COV-2 PT-RBD and eqACE2/Omicron BA.1-RBD complexes. a** Overall structure of the eqACE2/RaTG13-RBD, eqACE2/SARS-COV-2 PT-RBD and eqACE2/Omicron BA.1-RBD complexes. Boxes indicate the interaction patches. **b**, **c** Detailed interaction of the three complexes in Patch 1 (**b**) and Patch 2 (**c**). The complex structure is shown as cartoon and residues involved in the polar interaction are labeled, and H-bonds are shown as red dotted lines with a cutoff of 3.5 Å.

RBDs in complex with hACE2[21]. There are six substitutions on the interface of eqACE2 and hACE2 in complex with RaTG13-RBD (Supplementary Table 3). R357 of eqACE2 participates in the interaction with RaTG13-RBD, while R357 of hACE2 does not. Similarly, L79 of hACE2 takes part in RBD recognition whereas L79 of eqACE2 does not (Fig. 3f). The six substitutions of eqACE2/RaTG13-RBD also applies to equine and human ACE2 interacting with SARS-COV-2 PT-RBD (Supplementary Table 4). However, T82 of eqACE2 is not involved in RBD recognition, whereas its counterpart of hACE2 (M82) is (Fig. 3g).

hACE2 binds to SARS-CoV with significantly higher affinity than that of eqACE2. As the structure of eqACE2/SARS-CoV is yet to be determined, we labelled residues on hACE2 interacting with SARS-CoV and compared them with their counterparts on eqACE2. The same six substitutions as shown in hACE2/SARS-COV-2 PT-RBD and eqACE2/RaTG13-RBD also exist in hACE2/SARS-CoV-RBD (Fig. 3h), suggesting a similar mechanism behind the affinity variation.

We also analyzed the binding interface of eqACE2/Omicron BA.1-RBD. Interactions between Q325 of eqACE2 and F374 of Omicron BA.1-RBD can be observed, which locates away from the common binding interface between SARS-CoV-2 RBDs and ACE2 orthologs (Fig. 3i, j, Supplementary Table 5). Structure alignment showed that F374 of Omicron BA.1-RBD interacts with Q325 of eqACE2 (Supplementary Fig. 3d) and stretches the α2 helix of Omicron BA.1-RBD (Y365-N370, Supplementary Fig. 1b). Besides, the N417 of Omicron BA.1-RBD no longer participates in the binding with eqACE2 (Fig. 3k and Supplementary Table 6), which is consistent with the molecular interaction between hACE2 and Omicron BA.1-RBD (Fig. 3j)[34].

**The key residues for SARS-COV-2 PT-RBD and Omicron BA.1-RBD binding to eqACE2.** We next focused on the distinctive residues on RBDs from RaTG13 and Omicron BA.1

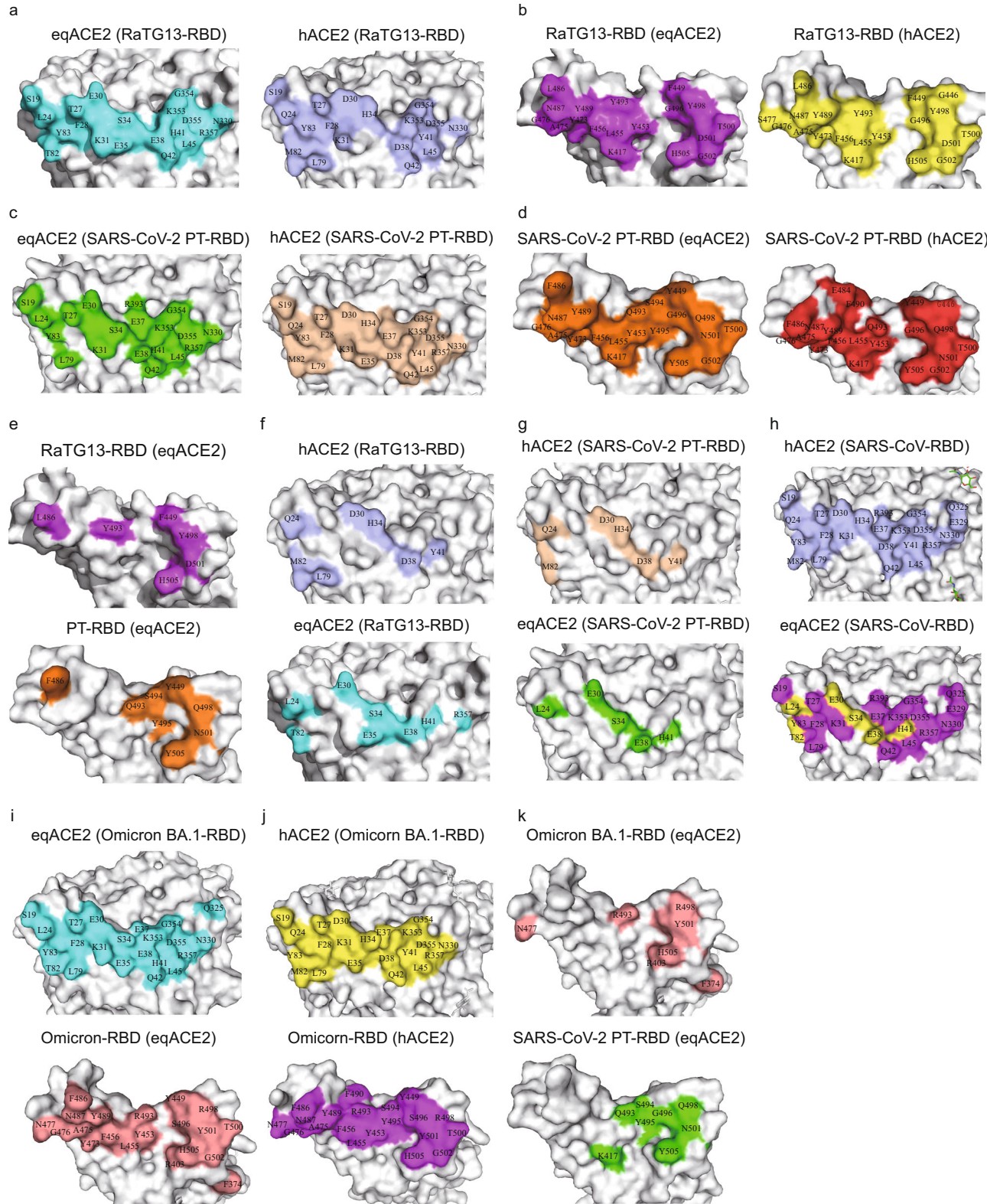

**Fig. 3 Interface comparison among ACE2 orthologs binding with RaTG13-RBD, SARS-COV-2 PT-RBD, SARS-CoV-RBD and Omicron BA.1-RBD.**
**a–d** Binding interface of eq (**a**, left) and hACE2 (**a**, right) with the RaTG13-RBD (**b**), and the corresponding interface of eq and hACE2 binding to the SARS-COV-2 PT-RBD (**c**, **d**). **e–g** Interface difference of the RBDs between the eqACE2/RaTG13-RBD and eqACE2/SARS-COV-2 PT-RBD complexes (**e**), ACE2s between the eqACE2/SARS-COV-2 PT-RBD and hACE2/SARS-COV-2 PT-RBD complexes (**f**), and ACE2s between the eqACE2/RaTG13-RBD and hACE2/RaTG13-RBD complexes (**g**). **h** Binding sites of hACE2 (up, light blue) and its corresponding residues in eqACE2 (purple, down). Particularly, substitutions of referred residues are labelled in yellow. **i**, **j** The binding interface of eqACE2/Omicron BA.1-RBD (**i**) and hACE2/Omicron BA.1-RBD (**j**) are shown and the interacting residues are labeled. **k** The distinctive interacting residues of RBDs between eqACE2/Omicron BA.1-RBD and eqACE2/SARS-COV-2 PT-RBD.

involved in binding to eqACE2. Six out of seven identified substitutions (F486L, Q493Y, Y449F, Q498Y, N501D, Y505H) between the SARS-CoV-2 RBD and RaTG13 RBD bound to eqACE2 have already been evaluated and analyzed in previous study[21] (Supplementary Fig. 4). Herein, we addressed the effect of the remaining one by constructing two mutants, R494S (which mutated R494 of RaTG13-RBD to its SARS-COV-2 PT-RBD counterpart S494) and 7-mutant (in which all seven substitutions were introduced into the RaTG13-RBD). Then, we prepared R494S, 6-mutant[21] and 7-mutant and conducted SPR analysis to measure their affinity to wild-type (wt) eqACE2 and hACE2. We found that the binding affinity of R494S for eqACE2 increased by ~3.6-fold, whereas no significant change was observed in the binding affinity with hACE2. Consistent with previous results, the affinity of the 6-mutant binding to eqACE2 or hACE2 increased to a similar level compared to that of the SARS-COV-2 PT-RBD. The 7-mutant, however, exhibited ~2-fold higher affinity than that of the SARS-COV-2 PT-RBD when binding to eqACE2 but ~3-fold weaker binding to hACE2 (Fig. 4a).

We then conducted structure comparisons to determine the molecular mechanism behind the binding effect of S494. When binding to the SARS-COV-2 PT-RBD, S494 forms two H-bonds with E38 on eqACE2 (Fig. 4b). However, in the eqACE2/RaTG13-RBD complex, E38 is drawn by Q42 and K353 on eqACE2 and Y498 on RaTG13-RBD, thus pointing away from R494. Meanwhile, R494 is bent by two H-bonds with a water molecule (Fig. 4c). When R494 is substituted by S494, E38 may be dragged toward S494, which results in stronger interactions between S494 and E38 (Fig. 4b and Supplementary Fig. 4b). We further investigated the structures of hACE2/RaTG13-RBD and hACE2/SARS-COV-2 PT-RBD. As the side chain of D38 on hACE2 is shorter than E38, it only interacts with Y498 when binding to the RaTG13-RBD (Fig. 4d). In complex with the SARS-COV-2 PT-RBD, D38 forms an H-bond with Y449 instead of S494 (Fig. 4e). This is consistent with the SPR results that R494S has little effect on affinity to hACE2.

Previous study found that 9 of the 15 substitutions carried by the Omicron BA.1-RBD are involved in the hACE2 binding[34], but only seven of them participate in the eqACE2 binding (Fig. 3k), namely K417N, S477N, Q493R, G496S, Q498R, N501Y and Y505H. Nevertheless, we constructed 9 SARS-COV-2 PT-RBD mutants, in which we substituted the residues of SARS-COV-2 PT-RBD with their Omicron BA.1 counterparts, and evaluated their affinity with eqACE2 (Fig. 4f). As expected, G446S and E484A demonstrated comparable affinity with eqACE2 compared to SARS-COV-2 PT-RBD. On the other hand, K417N, S477N and Q498R also displayed similar binding affinities with eqACE2, while Q493R, G496S, N501Y and Y505H showed extensively decreased binding (from ~6.2- to ~20.7-fold).

To elucidate the structural basis of the decreased affinities, we compared the structure details of Q493R, G496S, N501Y and Y505H. Q493R loses the H-bond between Q493 and K31 due to the repel of positive charge (Fig. 4g). The Y505 of SARS-COV-2 PT-RBD has longer side chain than the H505 of Omicron BA.1-RBD and carries a hydroxyl, which forms an interaction of 3.6 Å with E37, while H505 of Omicron BA.1-RBD does not. In addition, Y505 of SARS-COV-2 PT-RBD interacts with R393 whereas H505 does not (Fig. 4g). G496S mutant displayed significant decrease in binding affinity to eqACE2 (Supplementary Table 5). To explain the weakened binding, we modelled the G496S substitution in SARS-COV-2 PT-RBD using PyMOL and chose the rotamer with the lowest steric strain to analyze. The result showed that the S496 has significant steric strain with its surrounding atoms (Fig. 4h). These clashes might push the residues away and destroy the H-bond network around the water molecule (Fig. 4i). It's

noteworthy that N501Y was identified as an enhancing substitution when SARS-CoV-2-RBD binds to human, dog and mouse ACE2s[41,44,45]. When binding to hACE2, N501Y substitution brings about new favorable non-bonded interactions with Y41 and K353 such as π-π interaction and strengthens the receptor binding (Fig. 4i), which is extensively analyzed in previous studies[41]. However, when introduced the N501Y to the SARS-COV-2 PT-RBD, it decreased the binding affinity to eqACE2 (Fig. 4f). To elucidate the mechanism behind the undermining effect, we modelled the N501Y substitution in SARS-COV-2 PT-RBD with PyMOL and even the rotamer with lowest steric strain demonstrated significant clashes (Fig. 4i). Additionally, in the eqACE2 the residue on site 41 is a histidine instead of tyrosine, which results in fewer interactions between ACE2 and RBD.

**Distinctive binding sites in the eqACE2 or hACE2 in complex with the RaTG13-RBD and SARS-COV-2 PT-RBD.** To evaluate the role of substituted residues in eqACE2 or hACE2 in binding to the RaTG13-RBD and SARS-COV-2 PT-RBD, we constructed six eqACE2 mutants, namely L24Q, E30D, S34H, E38D, H41Y and T82M, on which the corresponding residues were changed to their counterparts in hACE2. We refolded these mutants as well as wt eqACE2 and hACE2 and measured their affinity for the RaTG13-RBD and SARS-COV-2 PT-RBD. We found that the binding affinity of the RaTG13-RBD to eqACE2 is ~2.6-fold greater than that of hACE2. Among the six mutants, E38D and T82M displayed no significant change in affinity compared to wt eqACE2, while the affinity of the L24Q and H41Y mutants decreased by ~2.6- and ~2.7-fold, respectively, to a level similar to hACE2 binding. The affinity of the E30D and S34H mutants decreased by ~1.8- and ~2.2-fold, respectively (Fig. 5a). As for the SARS-COV-2 PT-RBD, the affinity for eqACE2 was ~32.2-fold lower than that for hACE2. The E30D mutants bound to the SARS-COV-2 PT-RBD as tightly as wt eqACE2, and the affinities of the S34H and E38D mutants mildly increased by ~1.6- and ~1.3-fold, respectively. The L24Q, H41Y and T82M mutants bound to the SARS-COV-2 PT-RBD ~2.6-, ~6.5- and ~13.0-fold tighter than wt eqACE2 (Fig. 5a).

We aligned the α1 and α2 helices of eqACE2 and hACE2 in complex and analyzed the reason for these changes in affinity. When binding to the RaTG13-RBD, the salt bridge between H41 in the eqACE2 and D501 in the RBD is replaced by an H-bond between Y41 in the hACE2 and T500 in the RBD. E30 of eqACE2 forms an H-bond with K417, whereas D30 in the hACE2 forms only vdw contacts with the RBD. T82 in the eqACE2 forms 1 vdw contact with L486 in the RBD, which is similar to M82 in the hACE2 forming 2 vdw contacts. Although the sidechain of E38 in the eqACE2 is longer than that of D38 in the hACE2, their contacts are quite similar. As for L24 and S34, both substitutions interfere with the hydrophobic environment (Fig. 5b).

When binding to the SARS-COV-2 PT-RBD, Q24 and Y41 in the hACE2 introduced novel H-bonds into the interaction. Y489 and F486 of SARS-COV-2 PT-RBD form a small patch of hydrophobic interactions with F28, L79, M82 and Y83 of hACE2[16], whereas eqACE2 T82 is hydrophilic and sabotages the hydrophobic interaction. As a result, F486 of the SARS-COV-2 PT-RBD is released from the hydrophobic patch and further undermines the hydrophobic patch. Both E30 in the eqACE2 and D30 in the hACE2 form an H-bond with K417 (Fig. 5c). E38 in the eqACE2 and D38 in the hACE2 point toward separate directions, thus interacting with distinct clusters of RBD residues. S34 of eqACE2 forms an H-bond with K417 in the SARS-COV-2 PT-RBD, but H34 on hACE2 forms an H-bond with Y453 instead (Fig. 5c).

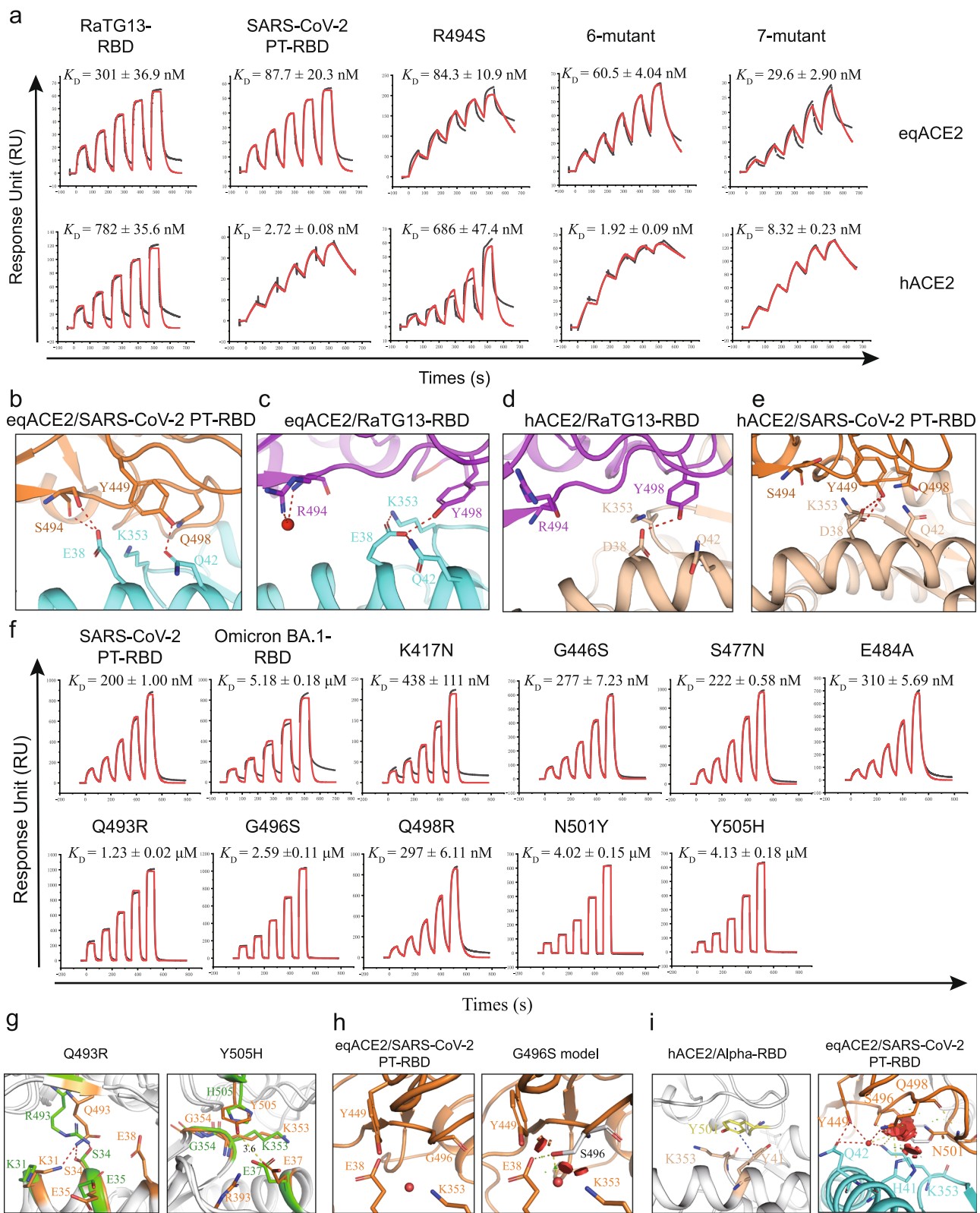

**Cross-reactive immunity of the SARS-COV-2 PT-RBD to the RaTG13-RBD.** Our previous work indicates that a cross-reactive immune response of SARS-CoV-2 to RaTG13 exists in humans[21,46,47]. To assess whether a similar circumstance applies to horses, we performed flow cytometry to examine whether a monoclonal antibody can block recognition between eqACE2 and the RaTG13-RBD or SARS-COV-2 PT-RBD. We chose CB6, which is reported to cross-neutralize RaTG13[21,48], to block the interaction between RaTG13-RBD/eqACE2, SARS-COV-2 PT-RBD/eqACE2 and RaTG13-RBD/hACE2. We found that events representing ACE2-expressing cells binding to RBDs were decreased in a CB6 concentration-dependent manner (Supplementary Fig. 5a, b).

**Fig. 4 Structural and functional analysis of key residues in the molecular interaction of eqACE2/PT-RBD and eqACE2/Omicron BA.1-RBD. a** SPR analysis of the binding affinity of wt/mutated RaTG13-RBD and SARS-COV-2 PT-RBD with human or eqACE2. **b**–**e** Structural details of site 494 of the RBD and involved residues of eqACE2/SARS-COV-2 PT-RBD (**b**), eqACE2/RaTG13-RBD (**c**), hACE2/RaTG13-RBD (**d**) and hACE2/SARS-COV-2 PT-RBD (**e**). RaTG13-RBD, SARS-COV-2 PT-RBD, eqACE2 and hACE2 are colored purple, orange, cyan, and wheat, respectively. **f** SPR analysis of the binding affinity of SARS-COV-2 PT-RBD, Omicron BA.1-RBD and mutated SARS-COV-2 PT-RBD. **g** The structural details of eqACE2/SARS-COV-2 PT-RBD (orange) and eqACE2/Omicron BA.1-RBD (green) around the sites referred above each picture. Residues involved in the interaction are represented as sticks. **h** The original conformation surrounding G496 of eqACE2/Omicron BA.1-RBD (left) and the modelled conformation of G496S (right) are shown. The modelled S496 is colored in white and the modelled H-bonds are represented by yellow dashes. The water molecule was represented as red sphere. The disks indicate pairwise overlap of atomic van der Waals radii between atoms. Small green disks and large red disks indicate slight and significant van der Waals overlap. Everything else lies between those extremes. **i** Structural comparison of Y501 and the involved residues of hACE2/Alpha-RBD (left) and modelled N501Y substitution of eqACE2/SARS-COV-2 PT-RBD. The water molecule is shown as red sphere and the modelled Y501 is colored in white. The π-π interaction, actual polar interaction and putative polar interaction for modelled Y501 are represented with blue, red and yellow dotted lines, respectively. The disks indicate pairwise overlap of atomic van der Waals radii between atoms. The dissociation constant ($K_D$) of each binding test is calculated from 3 independent repeats and are presented as mean ± SD. Source data are provided as a Source Data file.

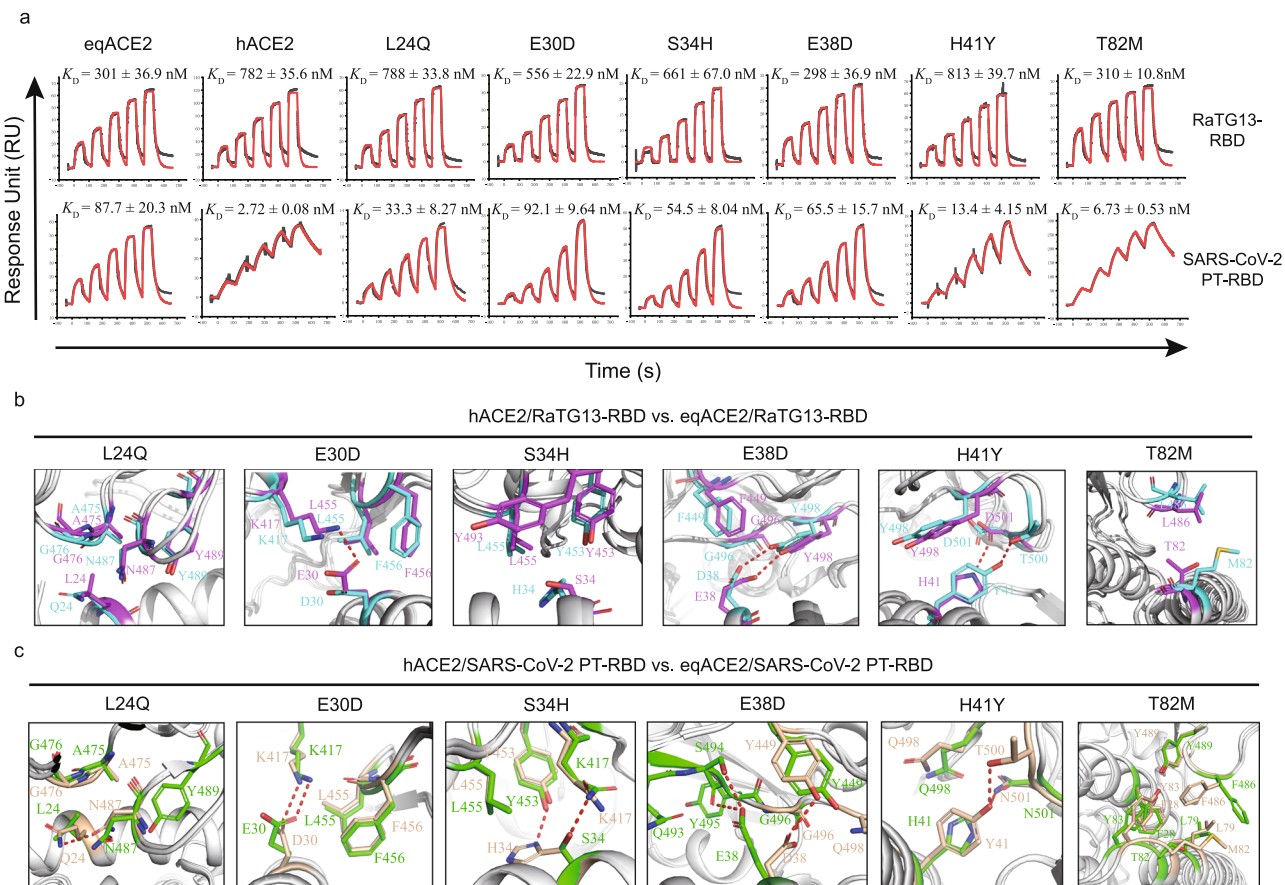

**Fig. 5 Structural analysis of ACE2 orthologs and the RaTG13-/SARS-COV-2 PT-RBD. a** SPR analysis of hACE2 and wt/mutated eqACE2 binding to the RaTG13-RBD or SARS-COV-2 PT-RBD. The dissociation constant ($K_D$) of each binding test is calculated from 3 independent repeats and are presented as mean ± SD. Source data are provided as a Source Data file. **b**, **c** Structural comparison between equine and hACE2 binding to the RaTG13-RBD (**b**) or SARS-COV-2 PT-RBD (**c**). Residue substitution of the eqACE2/RaTG13-RBD, hACE2/RaTG13-RBD, eqACE2/SARS-COV-2 PT-RBD and hACE2/SARS-COV-2 PT-RBD are colored in purple, cyan, green, and wheat, respectively. Involved residues are shown as sticks in corresponding color.

To explain the cross-immunity of CB6 against RaTG13-RBD, we labelled the overlapping residues of SARS-COV-2 PT-RBD bound by both eqACE2 and CB6[48] and compared their counterparts on RaTG13-RBD (Supplementary Fig. 5c, d). We found that 9 out of 13 overlapping residues are conserved between the SARS-COV-2 PT-RBD and RaTG13-RBD (Supplementary Fig. 5e), which may render CB6 capable of competitively blocking the RaTG13-RBD from binding to eqACE2.

## Discussion

Multiple studies indicate that SARS-CoV-2 may bind to eqACE2 with high affinity and use the receptor to efficiently invade eqACE2-expressing cells[18,19,49]. Furthermore, a similar high-affinity binding pattern has also been observed regarding SARS-CoV, two pangolin CoVs and RaTG13[19,21,40]. In this study, we found that eqACE2 can broadly recognize SARS-CoV, SARS-CoV-2, bat-origin CoV RaTG13 and pangolin-origin CoVs GX/

P2V/2017 and GD/1/2019 RBDs with high affinity. Thus far, there have been no reports on horses infected with SARS-CoV-2. Taken into account that eqACE2 is phylogenetically closely related to the ACE2s of bats, the well-known reservoir hosts of CoVs[19], the potential broad binding spectra of eqACE2 binding to SARS-CoV-2-related CoVs pose a severe threat of potential infection and virus spillover. Thus, horses should be closely monitored in case recombination events occur and eventually lead to interspecies transmission. It's also worth noting that previous studies have identified rabbits as another potential target for SARS-CoV-2 infection, whose ACE2 ortholog demonstrated high affinity to RBDs of SARS-CoV, SARS-CoV-2, RaTG13 and GD/1/2019[18,21] and efficiently support the pseudovirus infection of these CoVs[18]. Additionally, rabbit was also observed to be susceptible to SARS-CoV-2 experimental infection and can support productive replication of the virus[50]. Therefore, obtaining the structure of rabbit ACE2 and relevant CoVs are also important for the understanding of interspecies transmission.

Multiple key sites for receptor binding to the SARS-COV-2 PT-RBD have been identified as mutational hotspots and exert significant impact on host range[21,40,44,51], including residues 493, 498 and 501. In this study, we found that the S494 residue of the SARS-COV-2 PT-RBD plays an important role in binding to eqACE2. Point mutation revealed that R494S strengthens the RBD binding to eqACE2 but barely affects binding to hACE2.

Crystal structures of the eqACE2/RaTG13-RBD and eqACE2/ SARS-COV-2 PT-RBD complexes unveiled the molecular mechanism behind the high-affinity binding of eqACE2 to the two RBDs. Six substitutions were identified at the interface of equine or hACE2 binding to the RaTG13-RBD. Introduction of the H41Y and T82M mutations led to significantly increased affinity to the SARS-COV-2 PT-RBD. It is noteworthy that the RaTG13-RBD and SARS-COV-2 PT-RBD displayed opposite preference regarding the residue at site 41, where H41Y tightened binding to the SARS-COV-2 PT-RBD and is explained by a previous report but weakened binding to the RaTG13-RBD[39]. The reason may be the interacting residue 501 on the RBD, which is Asn in the SARS-COV-2 PT-RBD and Asp in the RaTG13-RBD. D501 of the RaTG13-RBD forms a salt bridge with H41 in the eqACE2, which strengthens the RBD interaction.

Previous work reported that the N501Y substitution enhanced SARS-CoV-2 RBD binding to human, dog and mouse ACE2s[41,44,45]. Herein, contrary to the influence on binding affinity with human and mouse ACE2, N501Y extensively weakens the affinity of SARS-CoV-2 RBD with eqACE2, which results from the destruction of the interaction network of K353 in eqACE2. This finding together with the decreased affinity of Omicron BA.1-RBD with eqACE2 serves as a reminder of the complexity of receptor recognition mechanism of SARS-CoV-2, and that the binding spectrum of SARS-CoV-2 variants can be altered. It further addresses the importance of evaluating the host range of VOCs to advise surveillance on potential hosts. Thus far, although there are several reports about the susceptibility of animal hosts to Omicron BA.1 variant[52,53], no systematic evaluation of Omicron BA.1's host range has been published. There is an urgent need for a comprehensive investigation on Omicron BA.1's host range both in laboratory and in natural settings.

Cross-immunity of antibodies targeting the SARS-COV-2 PT-RBD against the RaTG13-RBD was first reported in our previous study[21]. Here, our research showed that cross-immunity of CB6[48] also applies to binding to eqACE2, which may result from conserved antibody-interacting residues on the SARS-COV-2 PT-RBD and RaTG13-RBD[21,48]. This indicates that SARS-COV-2 PT-RBD-based vaccines may well protect horses from both SARS-CoV-2 and RaTG13 infection, which might be instructive for preventing co-infection and cross-species leakage in horses.

In summary, we found that eqACE2 broadly recognizes SARS-CoV, SARS-CoV-2, bat coronavirus RaTG13 and pangolin coronaviruses GX/P2V/2017 and GD/1/2019 RBDs with high affinity. eqACE2 may also bind to SARS-CoV-2 VOCs, but with comparable or decreased affinities. Crystal structures of eqACE2/ RaTG13-RBD, eqACE2/SARS-COV-2 PT-RBD and eqACE2/ Omicron BA.1-RBD were further determined, and S494 of SARS-COV-2 PT-RBD was identified as an important residue involved in the eqACE2and RBD interaction. Additionally, we found that N501Y, the commonly recognized enhancing mutation when bound to human and mouse ACE2, exerts a decreasing effect on binding affinity with eqACE2. On top of that, we demonstrated that cross-immunity of CB6, a SARS-COV-2 PT-RBD-targeted monoclonal antibody reported to cross-neutralize RaTG13 pseudovirus[34], also applies to eqACE2. Our work revealed the molecular basis for cross-species transmission and potential animal spread of SARS-CoV-2 and highlights the importance of continuous surveillance of SARS-CoV-2 and its related CoVs to prevent potential spillover of CoVs.

## Methods

**Gene cloning.** The full-length coding sequence of hACE2 and eqACE2 were synthesized and cloned into the pEGFP-N1 vector for flow cytometry[21]. The coding sequences of the RaTG13-RBD (residues 319-541, GenBank: QHR63300.2), SARS-COV-2 PT-RBD (residues 319-541, GISAID: EPI_ISL_402119), GX/P2V/ 2017-RBD (residues R319-F541, GISAID: EPI_ISL_410542), GD/1/2019-RBD (residues R319-F541, GISAID: EPI_ISL_410721), mutated RaTG13 RBDs (R494S, 6-mutant, and 7-mutant) and mutated SARS-CoV-2 PT-RBDs (K417N, G446S, E484A, S477N, Q493R, G496S, Q498R, N501Y and Y505H) were cloned into the pCAGGS vector. The coding sequenced of five VOC RBDs (residues 319-541, GISAID for Alpha: EPI_ISL_683466, Beta: EPI_ISL_678615, Gamma: EPI_ISL_833172, Delta: EPI_ISL_2020954, Omicron BA.1: EPI_ISL_6640916) are also cloned into pCAGGS vector. The hACE2 (residues 18-740, NCBI Reference Sequence: NP_001358344.1),eqACE2 (residues 19-615, NCBI Reference Sequence: NC_009175.3) and mutated eqACE2 (L24Q, E30D, S34H, E38D, H41Y, T82M) were cloned into pET21a.

**Protein expression and purification.** The CB6 antibody was expressed and purified from the culture supernatants of Expi293F cells using a Protein A affinity column (GE Healthcare) and further purified by gel filtration using a Superdex$^{TM}$ 200 10/300 GL column (GE Healthcare). Gel filtration data was collected using UNICORN 7.5. Purified proteins were stored in a buffer containing 20 mM Tris-HCl and 150 mM NaCl (pH 8.0). Proteins for SPR assays were transferred to PBST (1.8 mM $KH_2PO_4$, 10 mM $Na_2HPO_4$ (pH 7.4), 137 mM NaCl, 2.7 mM KCl and 0.05% (v/v) Tween 20) buffer.

The wt RaTG13-RBD, SARS-COV-2 PT-RBD, VOC RBDs, mutated RaTG13-RBD and mutated SARS-COV-2 PT-RBD cloned in pCAGGS were expressed in Expi293F cells. Cell culture supernatants were collected, filtered through 0.22 μm filters, and purified by His-Trap HP (GE Healthcare) and Superdex$^{TM}$ 200 Increase 10/300 GL (GE Healthcare) chromatography. Purified proteins were stored in protein buffer (20 mM Tris-HCl (pH 8.0) and 150 mM NaCl). Plasmids containing hACE2, eqACE2 and mutated eqACE2s were transformed into E. coli BL21 (DE3) cells and overexpressed as inclusion bodies under 1mM IPTG induction. The inclusion bodies were then dissolved by dissolution buffer (6 M Gua-HCl, 10% v/v glycerol, 50 mM Tris-HCl, 100 mM NaCl, 10 mM ethylenediaminetetraacetic acid (EDTA), pH 8.0) and refolded. Briefly, 5 ml of the solution (30 mg/ml) was added drop by drop to 2.5 L refolding buffer (100 mM Tris-HCl, 400 mM L-Arg-HCl, 2 mM EDTA, 5 mM glutathione (GSH), and 0.5 mM oxidized glutathione (GSSG), pH 8.0). After gently stirring for 8 h, we concentrated and exchanged proteins into a buffer containing 20 mM Tris-HCl (pH 8.0) and150 mM NaCl and purified using a Superdex$^{TM}$ 200 Increase 10/300 GL column (GE Healthcare).

**Flow cytometry assay.** The plasmids containing hACE2 or eqACE2 fused with eGFP were transfected into BHK-21 cells. A mixture containing SARS-COV-2 PT-RBD (5 μg/mL) or RaTG13-RBD (5 μg/mL) and CB6 antibody was pre-incubated at 4 °C for 1 h and then incubated with the BHK cells at 4 °C for 1 h. Subsequently, cells were washed with PBS thrice and stained with 1:1000 diluted APC mouse anti-His secondary antibody (Miltenyi Biotec, Cat# 130-119-820) for 1 h before being analyzed using a BD FACS Canto FlowCytometer (BD Biosciences). The data for all samples were collected using BD FACS Canto Diva 8.0.3 and analyzed using FlowJo 7.6 (TreeStar Inc., Ashland, OR, USA).

**SPR analysis.** The refolded hACE2, eqACE2 and mutated eqACE2 were transferred into PBST buffer (1.8 mM $KH_2PO_4$, 10 mM $Na_2HPO_4$ (pH 7.4), 137 mM NaCl, 2.7 mM KCl and 0.05% (v/v) Tween 20) and biotinylated with an NHS-LC-

LC-Biotin kit. Then, the biotinylated proteins are immobilized on flow cell 2 of a SA chip. Flow cell 1 was used as the negative control. Then, serially diluted wt or mutated RaTG13-RBD and SARS-COV-2 PT-RBD were flowed over the chip in PBST buffer. Binding affinities were measured using a BIAcore 8 K (GE Healthcare) at 25 °C in the single-cycle mode. Binding kinetics data were collected using BIAcore[TM] 8 K control software 3.0.12.15655 and analyzed with Biacore[TM] Insight software (GE healthcare) using a 1:1 Langmuir binding model. The SARS-COV-2 PT-RBD, RBDs from VOCs and mutated SARS-COV-2 PT-RBD were immobilized on CM5 chips and the following protocol of SPR assay is the same.

**Crystallization**. The sitting-drop method was used to obtain the high resolution RaTG13-RBD/eqACE2 and SARS-COV-2 PT-RBD/eqACE2 complex crystals. In detail, purified complex proteins were concentrated to 5 and 10 mg/mL. Then, 0.8 μL protein was mixed with 0.8 μL reservoir solution. The resulting solution was sealed and equilibrated against 100 μL of reservoir solution at 18 °C and 4 °C. High-resolution eqACE2/RaTG13-RBD complex crystals were grown in 0.2 M Lithium chloride, 0.1 M Tris 8.0 and 20% (w/v) PEG 6000, eqACE2/SARS-COV-2 PT-RBD complex crystals grew in 10% (w/v) PEG 1000, 10% (w/v) PEG 8000 and eqACE2/ Omicron BA.1-RBD grew in 0.2 M Sodium bromide, 0.1 M Bis-Tris propane 8.5, 20% (w/v) PEG 3350.

**Data collection and structure determination**. Reservoir solution supplemented with 20% (v/v) glycerol was prepared as cryo-protectant to freeze crystals. The crystals were picked with a mini loop and then soaked in this buffer for a few seconds. Subsequently, the crystals were exposed to liquid nitrogen for freezing. Diffraction data were collected at Shanghai Synchrotron Radiation Facility (SSRF) BL19U1. Datasets were processed with HKL2000 software[54]. The structure of two complexes were determined by the molecular replacement method using Phaser[55] with the previously reported complex structure of the SARS-COV-2 PT-RBD in complex with hACE2 (PDB: 6LZG). The atomic models were completed with Coot[56] and refined with phenix.refine in Phenix[55], and the stereochemical qualities of the final models were assessed with MolProbity[57]. Data collection, processing, and refinement statistics are summarized in Supplementary Table 1. All structural figures were generated using PyMOL software (https://pymol.org/2/).

**Statistics and reproducibility**. Binding Studies—$K_D$ values for SPR experiments were obtained with BIAcore 8 K Evaluation Software (GE Healthcare), using a 1:1 binding model. The values shown are the mean ± SD of three replicates. Flow cytometry analysis—All experiments were performed in biologically independent duplicate. One representative result is shown in Supplementary Fig. 5.

**Reporting summary**. Further information on research design is available in the Nature Research Reporting Summary linked to this article.

## Data availability

The data that support this study are available from the corresponding author upon reasonable request. The atomic coordinates for the crystal structures of the eqACE2/ RaTG13-RBD and eqACE2/SARS-COV-2 PT-RBD complexes have been deposited in the Protein Data Bank (www.rcsb.org) The accession numbers for eqACE2/RaTG13-RBD, eqACE2/RaTG13-RBD and eqACE2/Omicron BA.1-RBD are 7W6R, 7W6U and 7XBY, respectively. Source data is provided with this paper. Source data are provided with this paper.

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

## Acknowledgements

We are grateful to Yan Chai for his advice on structure determination and Pengcheng Han for his advice on experiment design. We also appreciate Xiaoqian Pan, and Yumin Meng for their help in conducting daily experiments. We are grateful to the Pathogenic Microbiology and Immunology Public Technology Service Center for its support for flow cytometry assays. We acknowledge the staff of beamline BL19U1 at the Shanghai Synchrotron Radiation Facility for assistance during data collection. We also thank Y. Chen, B. Zhou, and Z. Yang from the Institute of Biophysics, Chinese Academy of Sciences, for their technical support with SPR assays. This work was supported by the Strategic Priority Research Program of the Chinese Academy of Sciences (XDB29010202), the National Key Research and Development Program of China (2021YFC0863300) and the National Natural Science Foundation (32100752). Kefang Liu was supported by Young Elite Scientists Sponsorship Program by CAST (2021QNRC001) and the China Post-doctoral Science Foundation (2021M700161).

## Author contributions

X.Z., Q.W., K.L. and G.F.G initiated and designed the project. Z.X. and X.K. purified the proteins and grew the crystals. Z.X., L.L. and K.L. performed the SPR analyses. K.L. and Z.X. conducted the flow cytometry assays with the help of A.Z. and X.Z. P.H. and J.Q. collected the structural data and solved the structures. Z.X., K.L., X.K., P.H. and J.Q. analyzed the data and prepared the figures. Z.X., K.L., P.D., C.D. and G.F.G wrote and revised the manuscript.

## Competing interests

The authors declare no competing interests.
