## [Peer Review File · Nature Communications]

Binding and structural basis of equine ACE2 to RBDs from SARS-CoV, SARS-CoV-2 and related coronaviruses.Editorial Note: This manuscript has been previously reviewed at another journal that is not operating a transparent peer review scheme. This document only contains reviewer comments and rebuttal letters for versions considered at Nature Communications.

Reviewers' Comments:

Reviewer #1:

Remarks to the Author:

In this revised manuscript, the authors added kinetics studies on the bindings between equine ACE2 and RBDs of five SARS-CoV-2 VOCs. They further determined a crystal structure of eqACE2/Omicron-RBD complex. Then through mutagenesis and binding kinetics studies, they identified key residues in Omicron-RBD responsible for its decreased affinity to eqACE2. Overall, the revisions have added a significant amount of new data and have made some improvements to the manuscript. My specific comments as shown below.

Major comments

My previous major concern "the rationale of picking equine ACE2 over other ACE2 orthologs is not strong enough" has not been addressed. The authors argued that they picked equine ACE2 based on the findings from their previous study (Liu et al. 2021. Cell). They further argued that "we believe eqACE2/Omicron-RBD complex structure is more important than rabbit ACE2/SARS-CoV-2 RBD complex structure." Unfortunately, I could not agree with the authors here.

First, the rationale of choosing a specific target for investigation should be based on the findings or progresses in the entire field rather than any single study.

Second, I agree that investigations on Omicron are indeed of significance in general, and as mentioned by the authors (lines 73-74), "the identification of potential animal host of Omicron-RBD is urgently needed". However, the new Figure 1b clearly showed that all VOC RBDs, including the Omicron-RBD, have significantly lower binding affinity to eqACE2 than RT-RBD does. The significance of solving eqACE2/Omicron-RBD structure is therefore relatively weak.

I still think that the current manuscript could be significantly improved by including rabbit ACE2 into the study and performing detailed comparisons between equine and rabbit ACE2 for their binding to RBDs. Since the authors mentioned that they are trying to determine the complex structure of rabbit ACE2 with SARS-CoV-2 RBD now, I would encourage the authors to include the new structure data to the manuscript.

Minor comments

1. Figures 1a, 2a, 5a, and 6c: labeling "PT-RBD" as "SARS-CoV-2 PT-RBD" would increase the clarity.
2. Figures 3 and 4 should be combined as one figure to facilitate reading and comparison.

Reviewer #2:

Remarks to the Author:

Understanding interspecies transmission of SARS-CoV-2 and variants is important against the COVID-19 pandemic. In this study, the authors determined the crystal structures of RBDs of prototype-SARS-CoV-2, Omicron, and RaTG13 in complex with equine ACE2 (eqACE2). N501Y was shown to decrease the binding with eqACE2.

Specific comments:

1. Page 10, line 199: "... although shares the same sequence with PT-RBD, gets loosened and results in a extended loop where F374 locates (Fig. S3d)." Can the authors provide more explanation why the interaction is "loosened"?
2. Page 13, lines 255-256: "... the Y501 of Omicron-RBD forms a π -cation interaction, which destroys

the H-bond network of K353 observed in eqACE2/PT-RBD (Fig. 5i).” The “ π -cation” is supposed to mean “ π -cation”? From Fig. 5i, it seems that the K353 ϵ -amino group is pretty far away from the aromatic ring of Y501. Can the authors check if they can form a cation- π interaction? The reduced number of H-bonds seems largely owing to the lack of the water molecule. But it seems that the missing water molecule is due to the lower resolution of the eqACE2/Omicron RBD structure, where 0 water molecule was present in the structure (Table 1). In addition, Q493R also contributes to the loss of H-bonds (Figure 5i). Same for page 17, lines 354-355 “N501Y extensively weakens the affinity of SARS-CoV-2 RBD with eqACE2, which results from the destruction of the interaction network of K353 in eqACE2” which also attributed the affinity decrease of N501Y with eqACE2 to the disruption of the H-bond network.

3. Page 14-15, lines 295-300: “RBD whereas D30 of hACE1 flipped away. To explain this, we compared the interactions of D30 in both ... hACE2 H41 in the two complexes apply different orientation. As a result, in the hACE2/PT-RBD the nitrogen atom of H41 forms a salt bridge with D30 and drags it closer, which enables D30 to form an H-bond with K417 (Fig. S4c).” In Fig. S4b, it is not clear how D30 is flipped away – D30 in both structures seem to adopt similar orientation. In addition, the different orientations of H41 in the two structures can be a refinement artifact at this resolution. “Fig. S4c” is supposed to refer to “Fig. 4b” here?

4. It is not clear how the VOC RBDs are made. In page 20, lines 414-415, a list of mutated residues on RBD is shown, but not all Omicron RBD mutations are listed here.

Typos and grammar used in the manuscript can be improved. For example,

1. Page 10, line 199: “a extended loop” should be changed to “an extended loop”.
2. Page 12, line 251: “ratamer” should be changed to “rotamer”.
3. “ π -citation interaction” is used throughout the manuscript – is it supposed to mean “ π -cation interaction”?
4. Page 15, line 296: “hACE1” should be changed to “hACE2”.

Reviewer #1

In this revised manuscript, the authors added kinetics studies on the bindings between equine ACE2 and RBDs of five SARS-CoV-2 VOCs. They further determined a crystal structure of eqACE2/Omicron-RBD complex. Then through mutagenesis and binding kinetics studies, they identified key residues in Omicron-RBD responsible for its decreased affinity to eqACE2. Overall, the revisions have added a significant amount of new data and have made some improvements to the manuscript....

Response: Thank you for your positive comments.

Major comments:

1. My previous major concern “the rationale of picking equine ACE2 over other ACE2 orthologs is not strong enough” has not been addressed. The authors argued that they picked equine ACE2 based on the findings from their previous study (Liu et al. 2021. Cell). They further argued that “we believe eqACE2/Omicron-RBD complex structure is more important than rabbit ACE2/SARS-CoV-2 RBD complex structure.” Unfortunately, I could not agree with the authors here.

First, the rationale of choosing a specific target for investigation should be based on the findings or progresses in the entire field rather than any single study.

Response: Thanks for the reviewer’s suggestions. We agree that rabbit ACE2 is also an important object. We are trying to determine the complex structure of rabbit ACE2 with SARS-CoV-2 RBD, but unfortunately, we have not obtained the complex structure yet and will continue on this work.

2. Second, I agree that investigations on Omicron are indeed of significance in general, and as mentioned by the authors (lines 73-74), “the identification of potential animal host of Omicron-RBD is urgently needed”. However, the new

Figure 1b clearly showed that all VOC RBDs, including the Omicron-RBD, have significantly lower binding affinity to eqACE2 than RT-RBD does. The significance of solving eqACE2/Omicron-RBD structure is therefore relatively weak.

Response: Thank you for your comments. In previous studies, Alpha, Beta and Gamma variants increased binding affinity to hACE2 due to the N501Y mutation. In addition, N501Y mutation also enhances the binding affinity of RBD to human, dog and mouse ACE2s. Moreover, Omicron BA.1-RBD carries important mutations including Q483R, Q498R, N501Y, etc. These sites are known as mutation hotspots and host range determinants (Gao and Wang., China CDC Wkly, 2021, PMID: 34703641) and we are curious about their effect on equine ACE2 and the structural basis behind it. As far as we know, equine ACE2 is the first ACE2 that is reported with decreased binding affinity upon N501Y mutation RBD. So, determining the complex structure is helpful for us to understand the molecular mechanisms behind it, which is also relevant to the interspecies transmission mechanism.

3. I still think that the current manuscript could be significantly improved by including rabbit ACE2 into the study and performing detailed comparisons between equine and rabbit ACE2 for their binding to RBDs. Since the authors mentioned that they are trying to determine the complex structure of rabbit ACE2 with SARS-CoV-2 RBD now, I would encourage the authors to include the new structure data to the manuscript.

Response: Thank you for your suggestion. Although we are trying to determine the complex structure of rabbit ACE2 and SARS-CoV-2-RBD, the high-resolution crystals are yet to be collected. We will continue to work on it.

Minor comments

1. Figures 1a, 2a, 5a, and 6c: labeling “PT-RBD” as “SARS-CoV-2 PT-RBD” would increase the clarity.

Response: Thank you for your suggestions. We have revised the figures accordingly.

2. Figures 3 and 4 should be combined as one figure to facilitate reading and comparison.

Response: We agree with the reviewer's suggestion and combined Figures 3 and 4 together as current Figure 3.

Reviewer #2

Understanding interspecies transmission of SARS-CoV-2 and variants is important against the COVID-19 pandemic. In this study, the authors determined the crystal structures of RBDs of prototype-SARS-CoV-2, Omicron, and RaTG13 in complex with equine ACE2 (eqACE2). N501Y was shown to decrease the binding with eqACE2.

Response: Thank you for your positive comments.

Specific concerns:

1. Page 10, line 199: "... although shares the same sequence with PT-RBD, gets loosened and results in an extended loop where F374 locates (Fig. S3d)." Can the authors provide more explanation why the interaction is "loosened"?

Response: Thank you for your comments. In the eqACE2/Omicron-RBD, the F374 of Omicron BA.1-RBD forms interactions with Q325 of eqACE2, which stretches the $\alpha 2$ helix. We have redrawn Fig. S3d to demonstrate the interactions between F374 and Q325 and changed the description as "Structure alignment showed that F374 of Omicron BA.1-RBD interacts with Q325 of eqACE2 (Fig. S3d) and stretches the $\alpha 2$ helix of Omicron BA.1-RBD (Y365-N370, Fig. S1a and S3d)"

2. Page 13, lines 255-256: "... the Y501 of Omicron-RBD forms a π -cation interaction, which destroys the H-bond network of K353 observed in eqACE2/PT-RBD (Fig. 5i)." The " π -cation" is supposed to mean " π -cation"? From Fig. 5i, it seems that the K353 ϵ -amino group is pretty far away from the aromatic ring of Y501. Can the authors check if they can form a cation- π

interaction? The reduced number of H-bonds seems largely owing to the lack of the water molecule. But it seems that the missing water molecule is due to the lower resolution of the eqACE2/Omicron RBD structure, where 0 water molecule was present in the structure (Table 1). In addition, Q493R also contributes to the loss of H-bonds (Figure 5i). Same for page 17, lines 354-355 “N501Y extensively weakens the affinity of SARS-CoV-2 RBD with eqACE2, which results from the destruction of the interaction network of K353 in eqACE2” which also attributed the affinity decrease of N501Y with eqACE2 to the disruption of the H-bond network.

Response: Thank you for your comments. The " π -citation" is a typo error and means for " π -cation". We reanalyzed the structure and found that the interaction network in Patch 2 of hACE2/Omicron BA.1-RBD is largely changed because of G496S, Q498R, N501Y and H505Y. So, it is unreasonable to explain the individual effect of N501Y on receptor recognition of Omicron BA.1-RBD. Therefore, we revised the explanation. Briefly, when binding to hACE2, the enhancing effect of N501Y substitution is largely due to its favorable non-bonded interactions with Y41 and K353 of ACE2 (Han et al., Nat Commun, 2021, PMID: 34671049). However, in eqACE2 the residue on site 41 is a histidine (His, H) and no π - π interaction was formed. The N501Y substitution results in significant steric clash, which undermines the interaction between eqACE2 and the mutated PT-RBD. The current Fig. 4i has been revised accordingly.

3. Page 14-15, lines 295-300: “RBD whereas D30 of hACE1 flipped away. To explain this, we compared the interactions of D30 in both ... hACE2 H41 in the two complexes apply different orientation. As a result, in the hACE2/PT-RBD the nitrogen atom of H41 forms a salt bridge with D30 and drags it closer, which enables D30 to form an H-bond with K417 (Fig. S4c).” In Fig. S4b, it is not clear how D30 is flipped away – D30 in both structures seem to adopt similar orientation. In addition, the different orientations of H41 in the two structures can be a refinement artifact at this resolution. “Fig. S4c” is supposed to refer to “Fig. 4b” here?

Response: Thanks for your comments. The D30 flipping away was under comparison with eqACE2 E30 in the eqACE2/RaTG13-RBD (Fig. 5a). We agree that the slightly different orientation of H41 in two structures might be a refinement artifact. Thus, we deleted the original Fig. S4b and its related contents in case there might be any confusion.

4. It is not clear how the VOC RBDs are made. In page 20, lines 414-415, a list of mutated residues on RBD is shown, but not all Omicron RBD mutations are listed here.

Response: Thank you for your comments, the coding sequence and GISAID number of VOC RBDs and the vectors are added to the Methodology "Gene Cloning" section. The expression and purification protocol for VOC RBDs is also included in the "Protein expression and purification" section.

5. Typos and grammar used in the manuscript can be improved.

Response: Thanks for your comment. We have checked the manuscript and corrected all the typos and grammar mistakes to the best of our efforts.